# Replication confers β cell immaturity

Sapna Puri[1], Nilotpal Roy[1], Holger A. Russ [1,5], Laura Leonhardt[1], Esra K. French[2], Ritu Roy[3], Henrik Bengtsson[3], Donald K. Scott[4], Andrew F. Stewart[4] & Matthias Hebrok[1]

Pancreatic β cells are highly specialized to regulate systemic glucose levels by secreting insulin. In adults, increase in β-cell mass is limited due to brakes on cell replication. In contrast, proliferation is robust in neonatal β cells that are functionally immature as defined by a lower set point for glucose-stimulated insulin secretion. Here we show that β-cell proliferation and immaturity are linked by tuning expression of physiologically relevant, non-oncogenic levels of c-Myc. Adult β cells induced to replicate adopt gene expression and metabolic profiles resembling those of immature neonatal β that proliferate readily. We directly demonstrate that priming insulin-producing cells to enter the cell cycle promotes a functionally immature phenotype. We suggest that there exists a balance between mature functionality and the ability to expand, as the phenotypic state of the β cell reverts to a less functional one in response to proliferative cues.

---

[1] Diabetes Center, Department of Medicine, University of California, San Francisco, CA, USA. [2] Department of Endocrinology, Diabetes and Metabolism, University of Pittsburgh, Pittsburgh, PA, USA. [3] Helen Diller Family Comprehensive Cancer Center, University of California, San Francisco, CA, USA. [4] Diabetes, Obesity and Metabolism Institute, Icahn School of Medicine at Mount Sinai, New York, NY, USA. [5] Present address: Barbara Davis Center for Diabetes, University of Colorado, Anschutz Medical Campus, Denver, CO, USA. Correspondence and requests for materials should be addressed to M.H. (email: Matthias.Hebrok@ucsf.edu)

The adult pancreatic β cell is highly evolved to efficiently control glucose homeostasis, and loss of β-cell function leads to diabetes. Towards expanding existing pools of cells (either from cadaveric donors or differentiation of stem cells) for cell replacement therapies, efforts have been directed towards identifying factors (synthetic or biological) that can trigger β-cell replication. Such efforts have underscored the resistance of adult β cells to replication. In contrast, early postnatal stages in mice (the first days or weeks of life) and humans (< 5 years)[1] constitute a time of significant expansion of the β-cell pool. β cells immediately after birth, however, are functionally immature; immature β cells have higher basal insulin secretion, resulting in insulin secretion at low levels of glucose[2–4]. The temporal separation of mature, glucose-sensitive insulin secretion, and replicative potential has led to the speculation that there exists an inverse relation between the maturation state and the ability of the β cell to divide.

Despite compelling evidence that these two β-cell features are negatively correlated, it has been difficult to dissect the functional state of a β cell that is either undergoing replication, or is competent to divide, primarily due to the small fraction of cells that are actively in the replicative phase of the cell cycle even in neonatal stages. Recently, gene expression analysis in sorted, replicating β cells found that multiple genes involved in proliferation were upregulated[5], while genes involved in maintaining the β-cell state were not, explaining the relative reduction in gene expression of maturation markers. These observations raised the important question as to whether proliferation and maturity are mutually exclusive states in β cells. Understanding the mechanisms that control the balance between functional maturity and proliferative capacity should inform efforts to improve function in β cells derived from human embryonic stem cells (hESC) and human-derived induced pluripotent stem cells (hiPSC). It should also instruct efforts to manipulate β-cell proliferation in vivo in humans with small-molecule activators to prevent progression from glucose intolerance to type 2 diabetes.

To address the connection between proliferation and functional mature state, we manipulated the expression of c-Myc[6], a cell cycle regulator, in β cells. Immortalized rodent β-cell lines have high c-Myc, and depletion of the protein leads to proliferative arrest[7]. Furthermore, proliferative silence in human β cells can be overcome through the ectopic expression of c-Myc[7]. The transcription factor has thus emerged as a key regulator of β-cell proliferation at physiological and non-transformative levels. We demonstrate that an inverse relationship exists between replicative capacity and cellular function in the β cell by modulating c-Myc expression. Deletion of endogenous c-Myc in β cells in vivo reduces the proliferating pool of cells in postnatal stages. Conversely, stabilization of c-Myc in β cells in vivo not only promotes replication, but concomitantly diverts β cells towards an immature phenotype, mimicking β cell functionality soon after birth[8]. Increased expression of c-Myc in hESC-derived β cells promotes replication as well, providing a platform to test the role of regulators of replication in a human system.

## Results

**c-Myc activity plays a role in β-cell identity and function**. c-Myc drives replication in INS-1, a rodent β-cell line that expresses the glucose-sensing and insulin-secretory machinery, with reasonable insulin-secretion function[7]. Depletion or pharmacologic inhibition of c-Myc in INS-1 leads to reduced proliferation[7]. Based on the predicted inverse relation between proliferative capacity and β-cell maturity, we postulated that reduced cell cycle entry would result in improved insulin secretion. INS-1 cells were treated with 10058-F4 (Myci), an inhibitor that blocks c-Myc–Max interaction[9]. Compared to control, Myci treatment resulted in reduced cell density (Fig. 1a and Supplementary Fig. 1a) and increased expression of β-cell genes that confer mature features (Fig. 1b), suggesting that blocking c-Myc activity improves maturation at the expense of proliferative potential. Glucose-stimulated insulin secretion (GSIS) was significantly improved from twofold in control cells to 4.5-fold in Myci-treated cells, in part due to reduced insulin secretion in basal glucose, indicating that c-Myc activity negatively impacts cellular function (Fig. 1c, d).

c-Myc inhibition was also probed using a Myc-targeting siRNA (siMyc). A substantial reduction of c-Myc mRNA and protein upon transfection with siMyc (Supplementary Fig. 1b, c) led to reduced cell density (Fig. 1e and Supplementary Fig. 1d) and diminished expression of proliferation genes Ki67 and Pcna (Supplementary Fig. 1e), but increased expression of β-cell genes (Fig. 1f). siMyc cells showed reduced insulin release under basal conditions and a greater response at high glucose (Fig. 1g, h, 3.5-fold versus 1.75-fold), confirming inhibiting c-Myc in a β-cell line leads to reduced proliferation, increased expression of β-cell markers, and improved secretory capacity.

In rodents, β-cell proliferation is robust in early postnatal stages. To investigate if c-Myc plays a role in vivo during this time frame within β cells, we first quantified Myc protein in islets isolated from juvenile and adult animals. We found increased Myc protein expression in juvenile islets, along with increased mRNA expression of a cell cycle regulator Cyclin a (Ccna) (Fig. 1i, j and Supplementary Fig. 1f), providing correlative evidence of higher c-Myc in replication-competent β cells. Maturation markers Pdx1, Nkx6.1, and Neurod1 were significantly upregulated in adult islets, and Mafa and Ucn3 showed a trend towards increased expression (Fig. 1j).

Second, a β-cell-specific Cre line (Ins-Cre) was crossed with a c-Myc conditional knockout mouse line ($Myc^{flox/flox}$, denoted as $Myc^{-/-}$)[10]. At postnatal day 16, a time of significant expansion, animals with either one copy or both copies of c-Myc deleted had reduced BrdU-positive β cells (Fig. 1k). Adult knockout animals displayed a trend towards glucose intolerance and reduced replicative machinery (Supplementary Fig. 1g, h). These data provide direct evidence that c-Myc plays a role in normal β-cell expansion early in life.

**c-Myc induces proliferation in adult β cells**. Strategies that enhance proliferation in β cells have been explored as a potential avenue for cell replacement therapies. Our data suggested, however, that modifying the proliferative capacity of β cells impacts function. To test if increased proliferation in vivo leads to decreased function, we exploited an inducible transgenic mouse model that expresses c-Myc under the Insulin promoter (Ins-c-Myc). In this system, c-Myc is highly elevated by tamoxifen (TAM), resulting in apoptosis and diabetes[11–14]. Previous work has demonstrated leaky low-level c-Myc activity in the Ins-c-Myc mouse without TAM[11,15]. Nuclear c-Myc was detected in β cells in untreated adult Ins-c-Myc mice (Fig. 2a and Supplementary Fig. 2a, b) which also presented with a greater than fivefold increase in BrdU labeling index over controls (Fig. 2b, c). In addition, c-Myc-positive cells co-labeled with Ki67, demonstrating active replication (Fig. 2d). Untreated Ins-c-Myc mice displayed greater islet mass (Fig. 2e, f) that further increased over the lifetime of animals (Supplementary Fig. 2c, d). Cell cycle regulators activated by c-Myc[7] were significantly upregulated in Ins-c-Myc β cells (Fig. 2g and Supplementary Fig. 2e). Together, these data indicate low-level activity of c-Myc in Ins-c-Myc cells leads to β-cell proliferation.

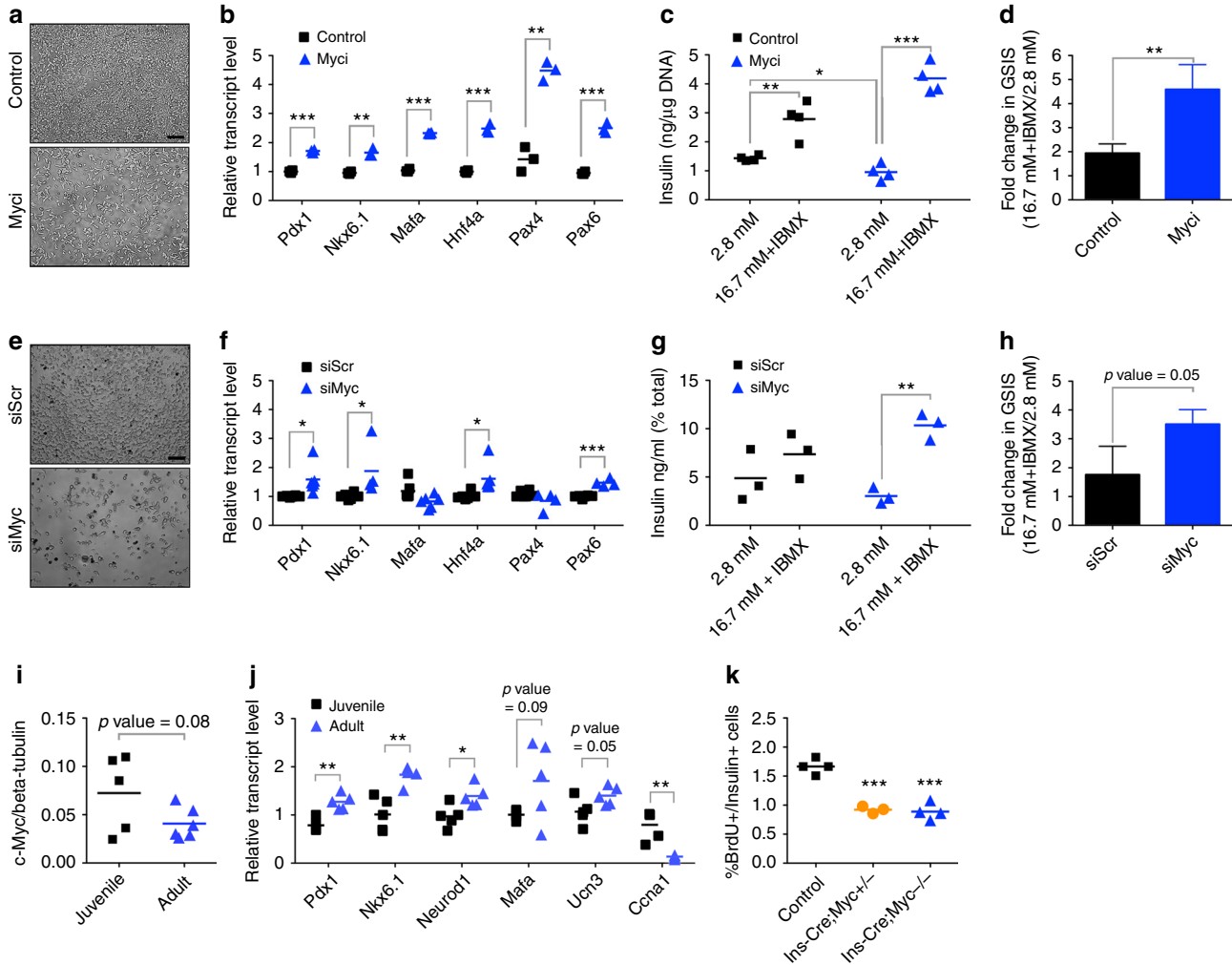

**Fig. 1** c-Myc plays a role in β-cell proliferation and function. **a** Cell density of INS-1 cells upon Myc inhibition with 40 μM Myci for 2 days as compared to control (DMSO) treatment. Scale bar, 100 μm. **b** Quantitative PCR to detect gene expression of key β-cell regulatory transcription factors upon Myc inhibition (Myci, 40 μM) after 2 days of inhibitor treatment, n = 3 per group. **p < 0.005, ***p < 0.0005, Student's t test. **c** Control (DMSO, n = 4) or Myci-treated (40 μM, 3 days, n = 4) INS-1 cells were subjected to glucose-stimulated insulin secretion (GSIS) under basal (2.8 mM glucose) followed by stimulatory (16.7 mM with 100 μM IBMX) conditions. *p < 0.05, **p < 0.005, ***p < 0.0005, Student's t test. **d** Fold change in GSIS in INS-1 cells treated with Myci (n = 4) as compared to controls (n = 4). Error bars indicate ± SD. **p < 0.005, Student's t test. **e** Cell density of INS-1 cells transfected with siMyc as compared to control samples transfected with a scrambled siRNA (siScr) for 5 days. Scale bar, 100 μm. **f** Quantitative PCR analysis of INS-1 cells to evaluate gene expression of several β-cell transcription factors upon reduction of c-Myc (siScr, n = 6, siMyc, n = 4–6). *p < 0.05, ***p < 0.0005, Student's t test. **g** Secretory response in INS-1 cells depleted of c-Myc (siMyc) as compared to control (siScr) cells. n = 3 per group. **p < 0.005, Student's t test. **h** GSIS measured in INS-1 cells with siMyc as compared to siScr samples. n = 3 per group. Error bars indicate ± SD, p = 0.05, Student's t test. **i** Western blot analysis of Myc levels in juvenile (3 weeks old, n = 5) islets versus adult (3 months old, n = 6) islets from wild-type mice. p = 0.08, Student's t test. **j** Quantitative PCR of β-cell maturation genes in adult islets (3 months old, n = 5) as compared to juvenile (3-weeks-old islets, n = 5) along with a cell cycle gene. *p < 0.05, **p < 0.005, Student's t test. **k** BrdU incorporation (expressed as %BrdU per Insulin + ve cells) in p16 pups to quantify actively replicating β cells in the transgenic (*Ins-Cre;Myc*+/−, n = 3 or *Ins-Cre;Myc*−/−, n = 4) animals as compared to control (n = 4) littermates. ***p < 0.0005, Student's t test

To test the effects of c-Myc on human β cells, c-Myc was delivered by adenovirus into hESC-derived insulin-producing cells[16]. Cells transduced with c-Myc had increased Ki67 staining as compared to control adenoviruses (Fig. 2h). These data emphasize the pro-proliferative effect of c-Myc in human β cells, strengthening the conclusion that c-Myc is a regulator of β-cell replication.

**β-cell proliferation leads to modified glucose regulation.** Higher replication and dysregulated insulin secretion are

hallmarks of juvenile β cells[2–4]. After establishing the pro-proliferative effect of c-Myc in β cells, we asked if c-Myc-induced proliferation affected β-cell maturity. Intraperitoneal glucose tolerance testing (IPGTT) revealed accelerated glucose clearance in *Ins-c-Myc* mice (Fig. 3a and Supplementary Fig. 3a). Although total β-cell mass was increased (Fig. 2e, f), there was no significant difference in islet insulin content (Fig. 3b). In vivo GSIS revealed a trend towards hyperinsulinemia under fasting conditions in transgenic animals (Fig. 3c, 0 min). In support of increased insulin levels under non-stimulatory conditions, random post-prandial glucose measurements indicated life-long hypoglycemia

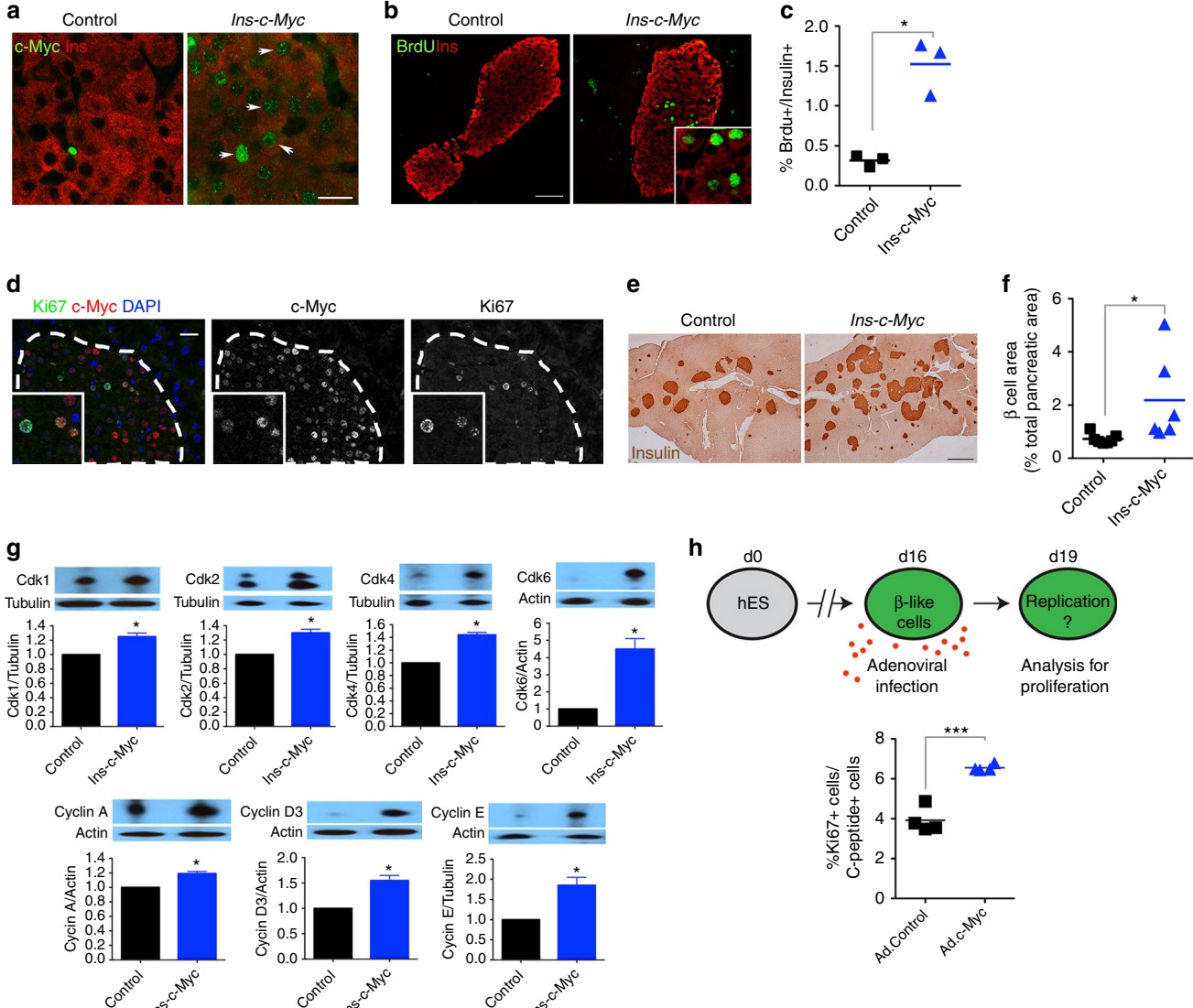

**Fig. 2** c-Myc stabilization increases β-cell replication. **a** Nuclear accumulation of c-Myc (green) in transgenic β cells (Insulin, red). Scale bar, 15 μm. Image shown is representative of at least three biological replicates. **b** BrdU (green) staining in *Ins-c-Myc* β cells (Insulin, red). Scale bar, 50 μm. **c** Quantification of BrdU incorporation in *Ins-c-Myc* animals ($n = 3$) as compared to controls ($n = 3$). *$p < 0.05$, Student's *t* test. **d** c-Myc expressing β cells co-stained with Ki67 in transgenic mice. Scale bar, 20 μm. Insulin staining (**e**) and islet mass quantification (**f**) in *Ins-c-Myc* mice ($n = 6$) as compared to controls ($n = 7$). Scale bar, 100 μm. *$p < 0.05$, Student's *t* test. **g** Quantification of the seven signature proteins in *Ins-c-Myc* islets. $n = 3$ per group, error bars indicate ± SD, *$p < 0.05$, Student's *t* test. All animals were three months old and were not administered TAM. **h** Cartoon summarizing experimental approach to test the effect of c-Myc on proliferation of hESC-derived β-like cells. Adenoviral infection was used to deliver either control (Ad. Control) or c-Myc (Ad.c-Myc) to hESC-derived β-like cells, and Ki67 staining quantified. $n = 4$. ***$p < 0.0005$, Student's *t* test

in *Ins-c-Myc* mice (Fig. 3d). Similarly, fasted blood glucose levels were markedly reduced in transgenic mice (Supplementary Fig. 3b). Pancreas weight was unchanged while body weight was reduced in *Ins-c-Myc* mice at 18 months (Supplementary Fig. 3c). To test function independently of systemic effects, isolated islets were challenged with glucose. Transgenic islets secreted high quantities of insulin at low glucose and had a stunted secretory response (Fig. 3e, f). Similar to what is observed in fetal and neonatal islets[17,18], *Ins-c-Myc* islets could mount a secretory response in the presence of glucagon, Glp-1 or forskolin (Fig. 3g). Overall, these data suggest that β cells poised to proliferate demonstrate high basal insulin secretion and an impaired secretory response.

**β cells in *Ins-c-Myc* islets have an immature phenotype.** A key feature of mature β cells is the presence of insulin-containing

secretory granules that at the ultra-structural level have a characteristic electron-dense core surrounded by a clear halo. While these features were clearly visible in control β cells (Fig. 4a), *Ins-c-Myc* β cells displayed increased proportion of immature secretory granules (less dense cores, with poorly defined halos) (Fig. 4a, b). Immature granules harbor proinsulin[19–21], an insulin precursor, and are typically localized to the peri-nuclear region. In line with the increased immature granules in *Ins-c-Myc* islets, proinsulin content was significantly increased (Fig. 4c), and the staining was modified to a diffuse cytoplasmic localization (Fig. 4d). Loss of processing enzymes that convert proinsulin to insulin leads to accumulation of immature proinsulin granules[19]. The processing enzyme prohormone convertase Pc1/3 (*Pcsk1*) had reduced expression in transgenic islets (Fig. 4e), suggesting that accumulation of proinsulin in *Ins-c-Myc* β cells resulted from reduced processing of proinsulin into mature insulin. Finally, GSIS

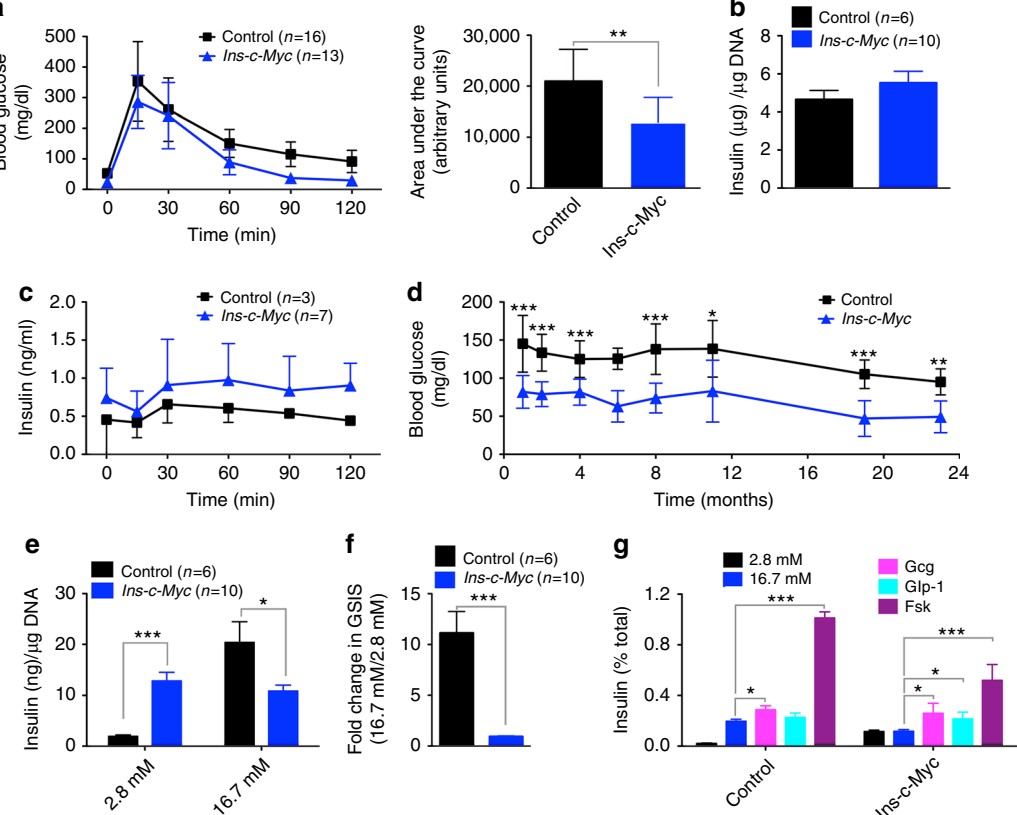

**Fig. 3** Imprecise glucose regulation in mice with stabilized c-Myc in β cells. **a** Glucose tolerance tests on three-month-old mice *Ins-c-Myc* animals (*n* = 13) as compared to littermate controls (*n* = 16). The corresponding area under the curve is noted. Error bars indicate ± SD. **p < 0.005, Student's *t* test. **b** Total insulin content in islets from transgenic (red bar, *n* = 10) and control (black bar, *n* = 6) animals. Error bars indicate ± SEM. **c** In vivo IPGTT on the transgenic (*n* = 7) and controls (*n* = 3) mice. Error bars indicate ± SD. **d** Measurement of glycemia under fed conditions in *Ins-c-Myc* and control animals. Numbers of control and transgenic animals analyzed at 1, 2, 4, 6, 8, 11, 19, and 23 months were: 22, 21; 27, 27; 22, 20; 6, 2; 5, 7; 5, 6; 17, 18; and 6, 5 respectively. Error bars indicate ± SD. *p ≤ 0.05, **p < 0.005, ***p < 0.0005, Student's *t* test. **e** In vitro glucose stimulated insulin secretion in islets from transgenic (red bars, *n* = 10) and control (black bars, *n* = 6) animals at basal (2.8 mM), and high (16.7 mM) glucose levels. Error bars indicate ± SEM. *p ≤ 0.05, ***p < 0.0005, Student's *t* test. **f** Stimulation index of insulin secretion in transgenic islets (red bar, *n* = 10) as compared to controls (black bar, *n* = 6). Error bars indicate ± SEM. ***p < 0.0005, Student's *t* test. **g** In vitro glucose stimulated insulin secretion in islets from 6-month-old transgenic (*n* = 3) and control (*n* = 3) in the presence of basal glucose (2.8 mM), high glucose (16.7 mM), glucagon (10 nM), Glp-1 (100 nM) or forskolin (100 μM). Error bars indicate ± SEM. *p ≤ 0.05, ***p < 0.0005, Student's *t* test

revealed increased proinsulin secretion at basal glucose from transgenic islets (Fig. 4f). Collectively, these observations indicate that c-Myc activity leads to immature insulin processing and packaging within the β cell.

mRNA analysis of β-cell genes in islets from adult mice revealed that although *Ins1* and *Ins2* were unchanged, *Pdx1*, *Nkx6.1*, *Mafa*, *Neurod1*, *Nkx2.2*, *Pax6*, and *Isl1* were reduced in the transgenic sample (Supplementary Fig. 4a). *Ucn3*, *Glut-2*, and the glucose sensor *Gck* were also reduced, pointing to diminished maturity. By immunostaining, Glut-2 was unchanged at three months in the transgenic mice but lost at later stages (Supplementary Fig. 4b). These data indicate that persistent replicative competence leads to progressive loss of β cell maturation. Chromatin immunoprecipitation revealed direct binding of Myc to canonical binding sites found in the upstream genomic sequences of key β-cell genes including *Pdx1*, *Pcsk1*, *Neurod1*, and *Ins2* (Fig. 4g). No canonical binding sites were found up to 10 kb upstream of *Nkx6.1*, *Mafa*, and *Ins1* genes.

As shown above, the features of β cells with elevated c-Myc overlap with those of cells in neonatal mice[2,22]: β cells proliferate, contain immature secretory granules, and exhibit high insulin secretion in low glucose[2,23]. To uncover mechanisms underlying such mutual exclusion between functional maturity and

replicative capacity, RNA sequencing was conducted on isolated *Ins-c-Myc* and control islets (Fig. 5a and Supplementary Fig. 5a). A nominal *p*-value cutoff at $10^{-6}$ revealed downregulation of 290 RNAs and upregulation of 175 RNAs in the *Ins-c-Myc* islets (Supplementary Data 1). A comparison of genes dysregulated in the *Ins-c-Myc* data set to published datasets of genes differentially expressed during β-cell maturation[2] revealed that approximately 30% of "immature" genes (upregulated in P1 islets) were upregulated in c-Myc islets (Supplementary Fig. 5b, c). Importantly, out of 81 transcripts upregulated during postnatal maturation, 74% were downregulated in the *Ins-c-Myc* dataset (Supplementary Fig. 5c). The above comparison suggests that genes associated with functional maturation during normal β-cell development are suppressed in the continued presence of c-Myc.

β-cell maturation involves not only increased expression of insulin-secretion genes but also the repression of disallowed genes that can lead to erroneous insulin secretion[24]. A subset of disallowed genes was upregulated in *Ins-c-Myc* islets, including *Slc16a1* (*Mct1*) (3.3-fold upregulated) and Hexokinase 3 (*Hk3*, 3.7 fold upregulated) (Supplementary Fig. 5d, e)[24,25]. In addition to genes influencing β-cell metabolism, mRNAs encoding cell cycle proteins were also upregulated in the transgenic samples (Supplementary Fig. 5f). These data provide evidence that c-

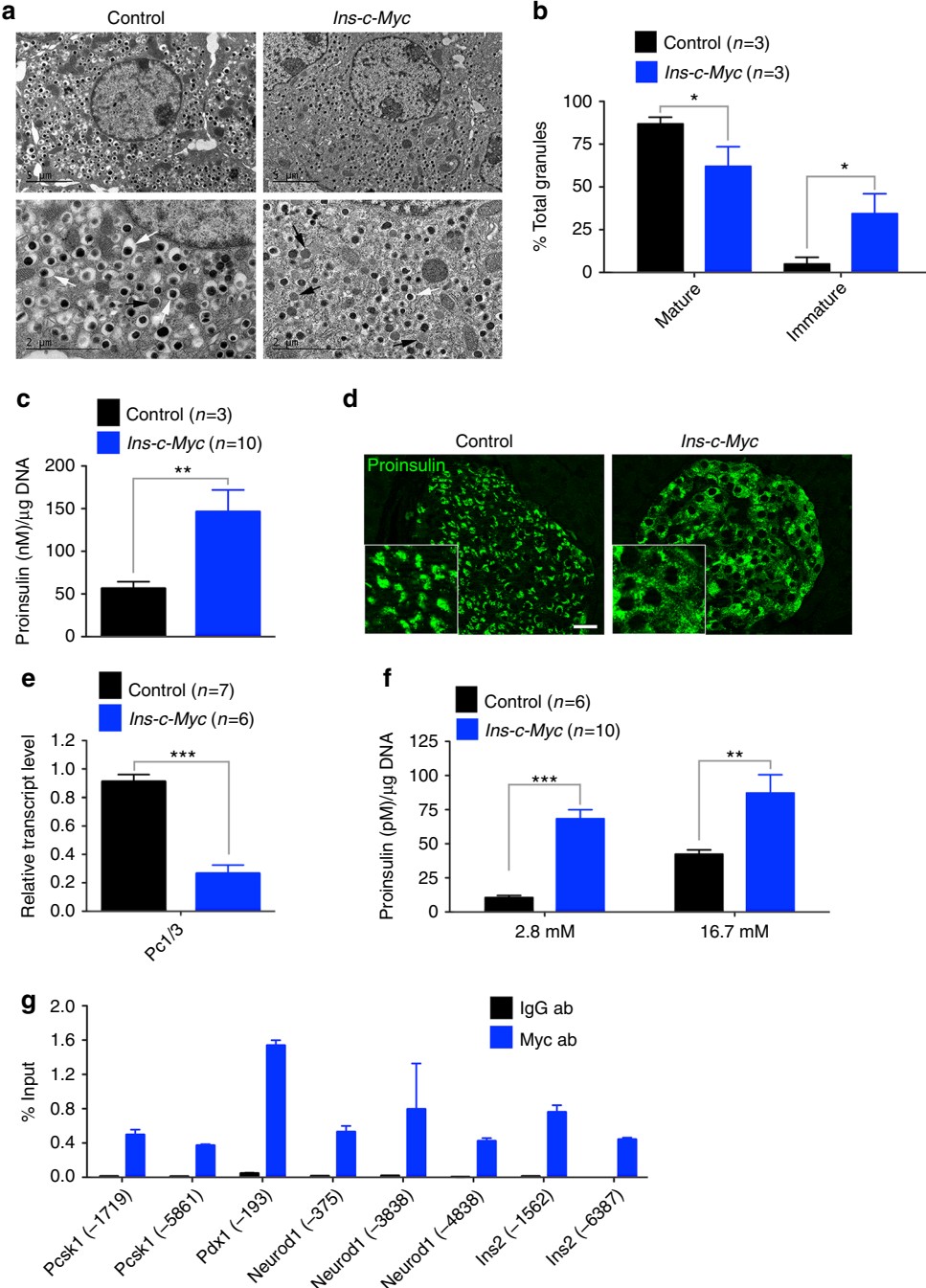

**Fig. 4** Accumulation of features of immaturity in c-Myc-expressing β cells. **a** Electron microscopy shows accumulation of classical mature insulin granules with a dark core surrounded by a clear halo (white arrows) in control islets. Immature granules have a less dense core and poorly defined halo (black arrows). Top panel, scale bar: 5 μm, lower panel, scale bar: 2 μm. **b** Quantification of mature and immature granules in control (black bars, $n = 3$) and transgenic (red bars, $n = 3$) samples. Error bars indicate ± SD. *$p < 0.05$, Student's $t$ test. **c** Total proinsulin content in islets isolated from control (black bar, $n = 6$) and transgenic (red bar, $n = 10$) animals. Error bars indicate ± SEM. **$p < 0.005$, Student's $t$ test. **d** Immunostaining for proinsulin reveals a diffuse distribution of proinsulin-containing vesicles in the transgenic islets. Scale bar: 20 μm. **e** Quantitative PCR to determine the level of *Pc1/3* in transgenic islets (red bar, $n = 6$) as compared to control islets (black bar, $n = 7$). Error bars indicate ± SD. ***$p < 0.0005$, Student's $t$ test. **f** Proinsulin secretion from transgenic (red bars, $n = 10$) versus control (black bars, $n = 6$) islets under basal (2.8 mM) and stimulatory (16.7 mM) glucose conditions. Error bars indicate ± SEM. **$p < 0.005$, ***$p < 0.0005$, Student's $t$ test. **g** Chromatin immunoprecipitation analysis to assess the recruitment of Myc to upstream canonical binding sites in the genes shown on *Ins-c-Myc* islets ($n = 2$). IgG was used as the negative control. Data are shown as percent of input. Error bars indicate ± SD

Myc-expressing β cells mimic gene expression profiles of neonatal, immature β cells.

**c-Myc causes a shift to a state poised for proliferation.** Gene ontology (GO) analysis of c-Myc islets revealed changes in

processes important for cell growth and proliferation, including upregulation of ribosomal proteins (Fig. 5b and Supplementary Fig. 5g). As expected, gene set enrichment analysis (GSEA) revealed an enrichment of Myc targets in transgenic islets (Fig. 5c). Several genes identified as Myc targets were involved in

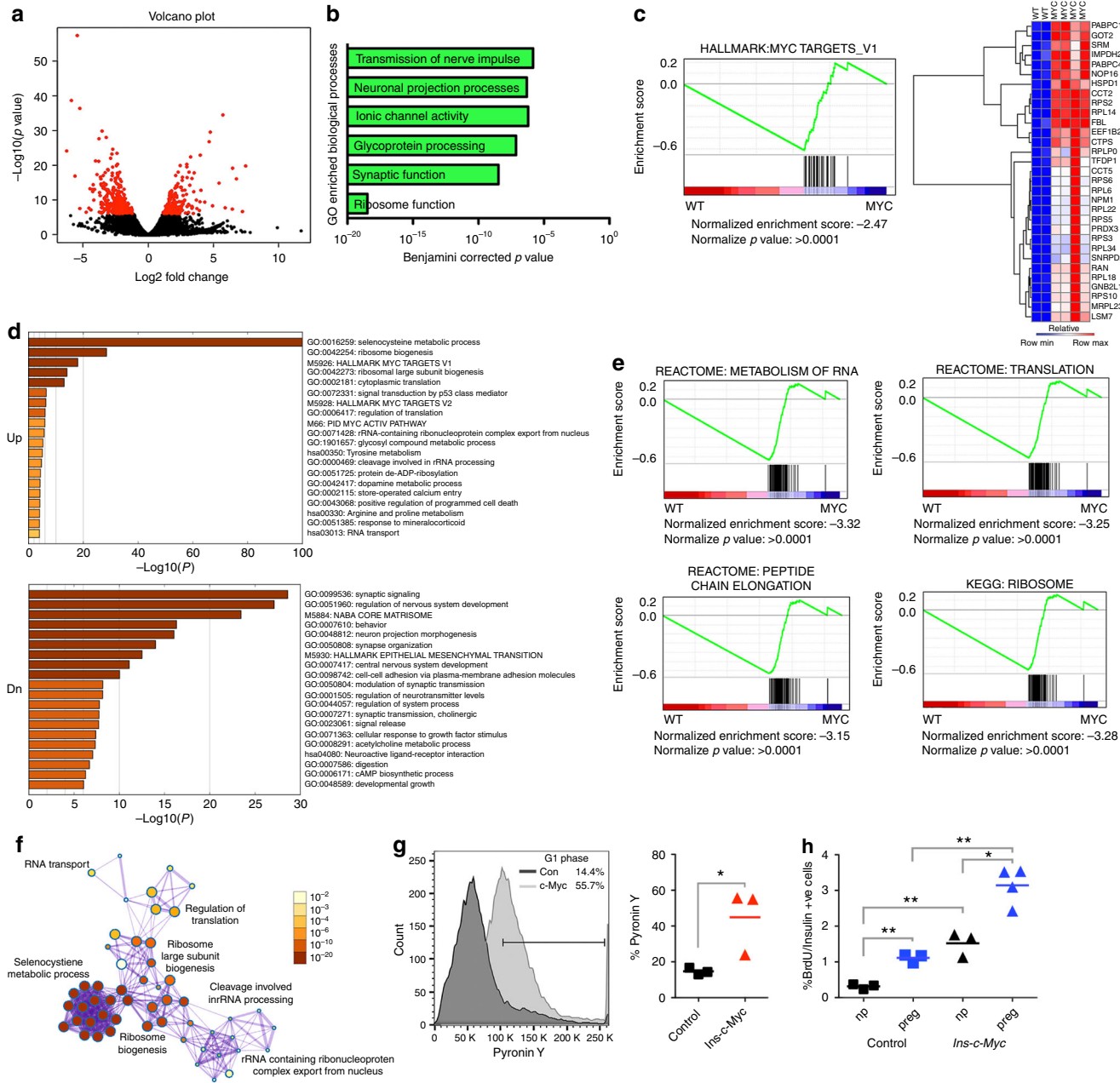

**Fig. 5** Global changes in β cells with stabilized c-Myc lead to an immature phenotype. **a** Volcano plot of transgenic (*Ins-c-Myc*) islets as compared to controls, with genes that are significantly (*p* value < 1e−06) changed marked in red. **b** Gene ontology analysis shows significant changes in cell growth processes, with the most significant change in "Ribosome Function" in transgenic islets. **c** Hallmark c-Myc targets that were highly significantly enriched in the transgenic samples identified using gene set enrichment analysis (GSEA). **d** Pathways differentially upregulated or downregulated in transgenic islets as compared to controls. **e** GSEA analysis of differentially expressed genes shows enrichment of genes involved in RNA and protein synthesis in the transgenic (myc) samples as compared to the controls (WT). **f** Metascape analysis shows significant overlap between RNA and protein synthesis pathways in the transgenic islet samples. Color-coding denotes *p* values. **g** Pyronin Y staining of islets isolated from *Ins-c-Myc* and control animals. *n* = 3 per group. \**p* < 0.05, Student's *t* test. **h** BrdU incorporation was quantified in control (squares) and *Ins-c-Myc* (triangles) animals at 3 months of age either not pregnant (black) or at 14.5 days of gestation (blue). *n* = 3 per group. \**p* < 0.05, \*\**p* < 0.005, Student's *t* test

RNA metabolism (*Pabpc1*, *Pabpc4*), mitochondrial function (*Got2*, *Hspd1*, *Prdx3*), biosynthetic reactions (*Srm*, *Impdh2*, *Ctps*, *Tfdp1*, *Ran*, *Gnb2l1*), protein folding (*Cct2*, *Cct5*), and ribosome biogenesis and function (*Nop16*, *Rps2*, *Rpl14*, *Fbl*, *Eef1b2*, *Rplp0*, *Rps6*, *Rpl6*, *Npm1*, *Rpl22*, *Rps5*, *Rps3*, *Rpl34*, *Snrpd2*, *Rpl18*, *Rps10*, *Mrpl23*).

Aside from direct Myc targets, other biosynthetic pathways significantly upregulated in transgenic samples included "Seleno-cysteine metabolic processes" (GO:0016259), "Ribosome

biogenesis" (GO:0042254), "Cytoplasmic translation" (GO:0002181), and "Regulation of translation" (GO:0006417), among others. Genes involved in "Synaptic signaling" (GO:0099536), "Regulation of nervous system development" (GO:0051960), and "Neuron projection morphogenesis" (GO:0048812), among others, were significantly downregulated (Fig. 5d). Thus, pathways encompassing genes involved in RNA and protein metabolism, and processes critical for cell replication were significantly enriched in transgenic islets (Fig. 5e, f). These

findings indicated that c-Myc in the β cell increases metabolic activity, including biosynthetic processes that are precursors to cell division. An upregulation of such biosynthetic pathways should result in increased levels of RNA within cells, which correlates with entry into the cell cycle (transition from G0 to G1)[26]. Increased total RNA was detected in transgenic islet cells (Fig. 5g).

In addition to early postnatal stages, another period of β-cell replication occurs during pregnancy[27–30]. We hypothesized that if c-Myc primes β cells for replication, proliferative cues during pregnancy would further increase the fraction of β cells entering the cell cycle. At 14.5 days of gestation, Ins-c-Myc pregnant animals had higher numbers of BrdU-positive β cells as compared to Ins-c-Myc non-pregnant and control pregnant age-matched animals (Fig. 5h). Ins-c-Myc pregnant animals had a ~tenfold increase in the fraction of β cells actively incorporating BrdU compared to non-pregnant control animals ($3.1 \pm 0.5\%$ versus $0.31 \pm 0.07\%$, $p$ value = 0.0002, Student's $t$ test) at the same age. Furthermore, Ins-c-Myc non-pregnant animals showed levels of proliferating β cells similar to those found in pregnant controls. These data indicate that increased c-Myc activity primes β cells to respond positively to physiological cues that stimulate cell cycle entry.

**Differential chromatin marks on Myc targets in human β cells.** In order to extend our findings to humans, we analyzed existing data to identify histone marks using chromatin immunoprecipitation (ChIP) on β cells from juvenile and adult donors (Supplementary Fig. 6a)[31]. If MYC is active in juvenile human β cells as in mouse, an enrichment of H3K4 trimethylation, an activating mark, is expected in the promoter regions of MYC. Analysis of the transcriptional start site of c-MYC revealed that H3K4 trimethylation (H3K4Me3) peaks were indeed sharper and more numerous in the juvenile donor sample, pointing to transcriptional activation over the adult sample (Fig. 6a). Using data from the murine GSEA analysis (Fig. 5c) and genes from literature, we analyzed the distribution of H3K4Me3 peaks in MYC target genes (Fig. 6b, c and Supplementary Fig. 6b). Genes highlighted in purple (48% of the genes examined) showed differential H3K4Me3 around the transcriptional start site. In other words, genes positively regulated by MYC had increased activating H3K4Me3 chromatin marks in the juvenile versus adult sample, while the opposite was true for genes downregulated by MYC. A subset of the target genes (RCC1, ODC1) had increased H3K4Me3 and increased H3K27 acetylation in the juvenile sample, suggesting activation of enhancer elements (Fig. 6d). In addition to canonical MYC target genes, the disallowed genes Hk3 and Mct1 also showed an enrichment of H3K4Me3 in the juvenile sample (Supplementary Fig. 6b). The analysis of chromosomal signatures therefore suggests MYC target activation and functional immaturity in juvenile murine and human samples alike (Fig. 6e).

**Discussion**
It is well known that replicative capacity of β cells diminishes with age in humans and rodents[32–36]. The inverse correlation between proliferation and functional maturation in β cells has, however, been difficult to prove, due to the small fraction of adult replicating cells. We demonstrate that priming β cells to replicate leads to a compromised functional state, and c-Myc serves as an inverse dual regulator of β-cell proliferation and maturity (Fig. 6e). The Ins-c-Myc mouse model with ectopic activity of c-Myc generates a large pool of β cells that can potentially enter the cell cycle, and serves as a tool to interrogate the role of replication in quiescent β cells.

Changes in gene expression revealed increases in the biosynthetic machinery, cell cycle components and expression of replication markers, indicating that cells were primed for proliferation. The GO analysis of the Ins-c-Myc β cells revealed that the most significantly upregulated biological process included ribosome biogenesis and function. One property of neonatal cells is increased protein synthesis under basal glucose conditions[4]. Cells that are poised for division increase bio-energetic functions in order to expand the resource pool for replication, and c-Myc is a key regulator of these processes[6]. Total RNA content increase in c-Myc cells further suggested a shift from G0 to a G1 state, poised for entry into the cell cycle. These observations were in line with the observations in juvenile β cells, where a greater fraction of cells is in the cell cycle. c-Myc expression was elevated in juvenile samples, providing correlative evidence for the protein. Genetic depletion of c-Myc led to a ~50% reduction in the proliferative pool during postnatal expansion, demonstrating a role for this protein in the replicative phase of β-cell development.

As another example, during gestation in rodent models, β cells enter the cell cycle in response to proliferative cues in order to compensate for metabolic demands of the female[27–30,37]. What is significant is that during pregnancy in mice, an increase in β-cell mass is accompanied by increased c-Myc expression[29]. Several of the same cell cycle regulators that we see upregulated in the Ins-c-Myc β cells, namely cyclins A, D3, E, and cdk4, are also increased during pregnancy[29]. Thus, c-Myc could be playing a critical role in the β cell expansion that occurs during pregnancy. Our data demonstrate that in the presence of a physiological stimulus such as pregnancy, c-Myc β cells further increase the pool of replicating cells, supporting the idea that cells are poised to replicate and in a permissive environment are more likely to expand.

Examination of the functional state of β cells with c-Myc revealed numerous observations—β cells secreted high insulin in low glucose leading to hypoglycemia, exhibited poor GSIS, and had compromised expression of important function-maintaining genes, closely resembling an immature functional state. Furthermore, disallowed genes Mct1 and Hk3 were significantly upregulated. Overexpression of Mct1, a monocarboxylate transporter, in β cells leads to increased pyruvate-induced insulin secretion[38]. The disallowed hexokinases (hexokinase I, II, and III) are low Km enzymes that induce high insulin secretion under basal conditions. The primary glucose sensor in β cells, glucokinase (hexokinase IV), a high Km enzyme that responds to elevated levels of glucose to facilitate appropriate insulin secretion, was reduced (Supplementary Fig. 4a). Combined effects of increased glucose-sensing enzymes and Mct1 could lead to hyperinsulinemia in low glucose.

Several of the changes that the β cell undergoes upon expression of c-Myc mimic what is seen in dedifferentiated β cells—reduced expression of canonical transcription factors including Pdx1, Nkx6.1, and Mafa, upregulation of disallowed genes such as Mct1 and Hk3, and reduced expression of the glucose sensors Glut-2 and Gck. Unsurprisingly, there are overlaps between immature and dedifferentiated phenotypes at the transcriptome level[39–41]. The functional outputs of these phenomena differ, however, as immature β cells have poor control of the secretory machinery in the absence of a bolus of glucose, resulting in systemic hypoglycemia, as opposed to dedifferentiation, which eventually leads to diabetes due to severe stunting of either insulin expression or secretion in the presence of high glucose. Thus, immaturity and dedifferentiation present two related, yet distinct, facets of β cell dysfunction.

Aberrant proinsulin processing and dense core secretory granule morphology illustrate other examples of the c-Myc-induced immature phenotype. The expression of the proinsulin-processing enzyme Pc1/3 is low early in postnatal life[42], which

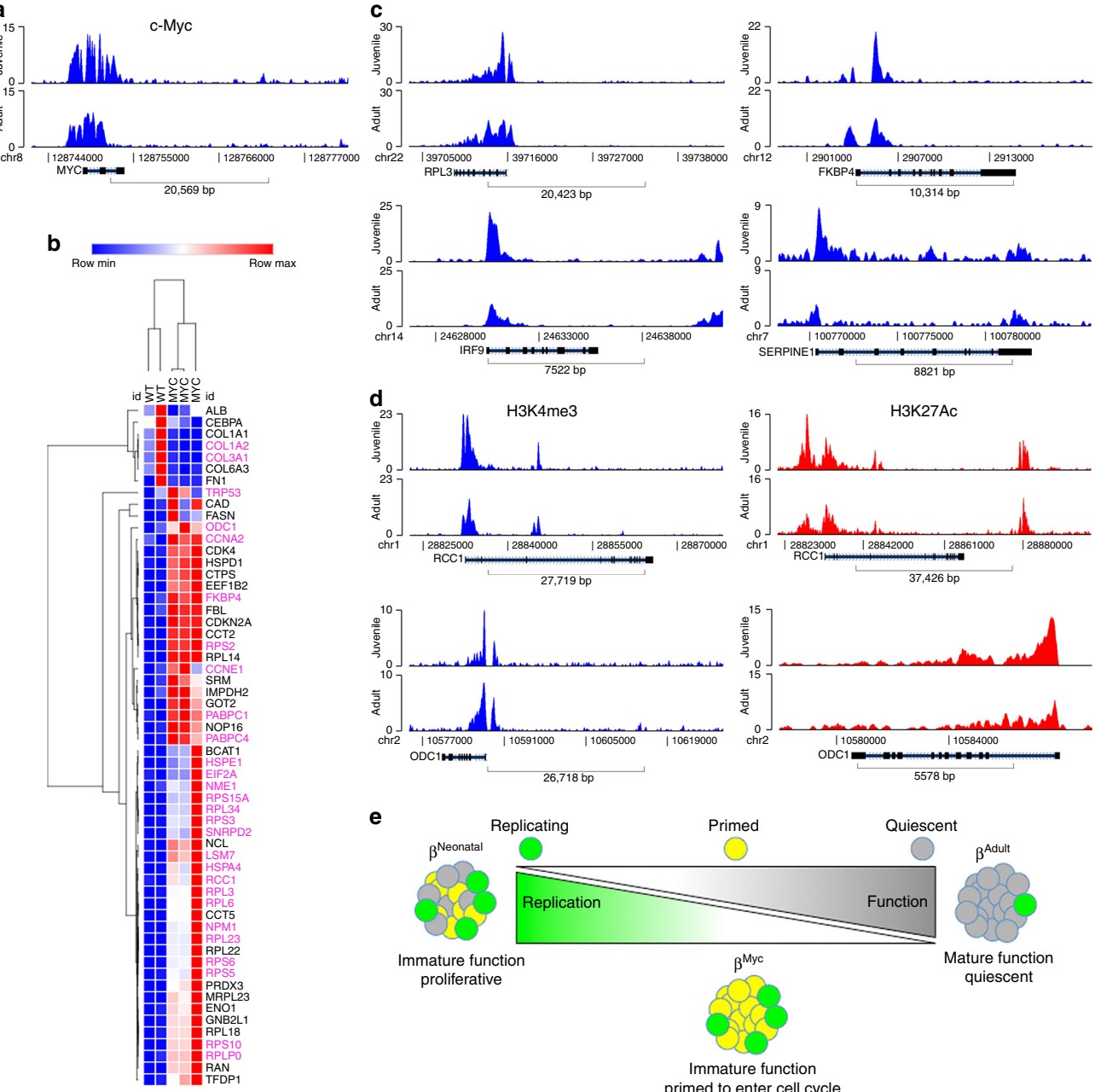

**Fig. 6** Analysis of human ChIP data reveals activated marks on c-Myc targets in juvenile samples. **a** H3K4Me3 ChIP-peak distribution across the c-Myc promoter in β cells isolated from a juvenile (5 years) and an adult (48 years) donor. **b** RNA-seq data demonstrating levels of several c-Myc targets in islets from *Ins-c-Myc* and control animals. Genes marked in purple show differential H3K4 trimethylation marks in the juvenile donor samples as compared to the adult sample. **c** H3K4Me3 ChIP-peak distribution in *RPL3*, *IRF9*, *FKBP4*, and *SERPINE1* genes in a juvenile (5 years) and adult (48 years) sample. **d** H3K4Me3 and HeK27Ac marks in the juvenile (5 years and 0.8 years respectively) samples as compared to the adult (48 years and 66 years respectively) samples of c-Myc target genes *RCC1* and *ODC1*. **e** Schematic representing a key role of c-Myc is the early stages of life, at a time of increased proliferation and immature function within β cells. With age, a reduction in c-Myc occurs concomitant with acquisition of maturation features and loss of replicative capacity. In the *Ins-c-Myc* animals, β cells continue to express c-Myc well into adulthood and throughout life, leading to a persistence of replicative capacity, and a failure of the cells to undergo maturation, thus leading to dysregulated glucose levels

results in the higher proportion of immature secretory granules present soon after birth. Adult *Ins-c-Myc* islets had reduced expression of Pc1/3, which may contribute to the accumulation of proinsulin and immature secretory granules, reminiscent of early postnatal β cells (Fig. 4)[21,43].

In vitro, the bidirectional switch between functionally immature, proliferative β cells and mature, fully functional, non-proliferative β cells is reversible. Rapidly replicating INS-1 cells

increased expression of β-cell regulators, demonstrated reduced basal insulin secretion, and had improved GSIS upon c-Myc silencing. Such an inverse relationship between proliferation and functional competence has been suggested for a human β-cell line as well[44]. c-Myc promoted replication in human ES-derived β cells, supporting the conclusions from the rodent models. A role for c-MYC in β-cell maturation and proliferation in humans is unknown, although analysis of the active chromatin marks on

genomes of juvenile and adult individuals suggested that MYC activity was increased at younger ages. Future work will define the exact role MYC plays in human β-cell function and expansion.

In conclusion, this study highlights a balance within the β cell —the ability to replicate compromises function. A transient loss of function is permissible if a small percentage of cells replicate, as occurs in adult islets. When a large fraction of β cells, however, is poised to divide, and cellular bioenergetics favor growth, overall β cell function deteriorates, leading to dysregulated insulin processing and release. In therapeutic terms, these findings suggest that agents activating human β-cell replication can be useful, as long as such activity is reversible.

## Methods

**Animals, glucose tolerance test, and hormone measurement.** Ins-c-MycER[TAM] (Ins-c-Myc) and the Myc[flox/flox] mice were obtained from Dr. Gerard Evans and Dr. Zhonghui Guan respectively. Physiological analyses including IPGTT were carried out on three-month-old male and female Ins-c-Myc mice unless otherwise noted. Only males were used for IPGTT analyses in three-month-old Ins-Cre;Myc[flox/flox] and Ins-Cre;Myc[flox/+] animals. After a 16–18 h fast, mice were weighed, blood glucose level measured using the Contour Glucometer, and injected intraperitoneally with a 1 M glucose solution at 10 μl per g body weight. Blood glucose was measured every 30 min for 2 h after injection. For in vivo hormone measurement, blood was collected either before and after a glucose challenge from the tail vein, spun down to collect serum, and stored at −80 °C with protease inhibitors (Roche). Insulin concentration was calculated using the Insulin EIA kit (ALPCO). All transgenic mouse experiments were performed in the absence of TAM. All animals were maintained in the barrier facility according to protocols approved by the Committee on Animal Research at the University of California, San Francisco.

**Glucose-stimulated insulin secretion.** Ten-size-matched islets (control or transgenic) were incubated in KRB with 2.8 mM glucose for 30 min with gentle shaking. The supernatant was discarded and islets incubated in KRB with 2.8 mM for 1 h. The supernatant was collected and frozen at −20 °C with protease inhibitors. Islets were subsequently incubated with 16.7 mM glucose for 1 h followed by 16.7 mM glucose and either 10 nM glucagon (Sigma), 100 nM Glp-1 or 10 μM foskolin (Sigma) for another hour. Supernatants were collected and stored at −20 °C with protease inhibitors. Total insulin was extracted overnight in acid/alcohol buffer, followed by DNA extraction[45]. For insulin secretion from INS-1 cells[46], 5 days after transfection, cells were treated as follows- cells were rinsed with KRB without glucose, and incubated with KRB + 2.8 mM (low) glucose for 2 h. Following the pre-incubation with low glucose, cells were incubated with 2.8 mM glucose in KRB for 2 h, and the supernatant collected for insulin quantification. Subsequently, cells were incubated with high (16.7 mM + 100 μM IBMX) glucose for another 2 h, and the supernatant collected. At the end of the high glucose incubation, acid/ethanol extraction was carried out overnight at 4 °C for quantification of total insulin levels. Total DNA was extracted from the cells for normalization of secretory capacity. Measurement of insulin was carried out using the Insulin Rodent Chemiluminescence ELISA kit (ALPCO). Proinsulin levels were detected using the Mouse Proinsulin ELISA kit (ALPCO).

**Cell culture and cell density quantification.** INS-1 cells (a kind gift from Dr. Chris Newgard) were grown in RPMI-1640 medium supplemented with 10% fetal calf serum, 10 mM HEPES, 2 mM L-glutamine, 1 mM sodium pyruvate and 0.05 mM 2-mercaptoethanol[46]. Cells were grown in 6 wells plates and transfected with siRNA (AllStars Negative Control siRNA, Qiagen, or siRNA against rat c-Myc, Dharmacon) using Lipofectamine 2000. INS-1 cells were treated with 40 μM 10058-F4 (Sigma) for the time indicated. Cell density was quantified by Cell Titer Glo Viability assay (Promega) and DNA quantification was carried out using the Quant-iT PicoGreen dsDNA Assay Kit (Life Technologies).

**Histology, immunofluorescence, and islet size quantification.** Pancreata were fixed in Z-Fix (Anatech) for 12–16 h at 4 °C and processed for paraffin embedding. Pancreatic sections were de-paraffinized, rehydrated and subjected to antigen retrieval by boiling in a water bath in Citrate buffer for 5 min, followed by no boiling for 30 s, and another 3 min of boiling. After cooling to room temperature, slides were washed in water followed by PBS, and incubated in blocking reagent (1% BSA in PBS) for 30 min, followed by incubation with the appropriate primary antibodies in the blocking reagent overnight at 4 °C. Slides were washed three times in PBS for 5 min each, followed by incubation for 30 min at room temperature in the appropriate secondary antibodies in blocking reagent. For immunofluorescence, slides were washed in PBS (three times for 5 min each) and mounted using VectaShield HardSet mounting medium with DAPI (Vector Laboratories)[47]. For immunohistochemistry, slides were further incubated for 30 min in ABC solution (Vectorlabs), washed in PBS, and developed using DAB reagent (Vectorlabs) as per manufacturer's instructions. The primary antibodies used were:

rabbit anti-c-Myc, 1:200 (#5605, Cell Signaling); mouse anti-Insulin, 1:500 (I2018, Sigma); guinea pig anti-Insulin, 1:500 (#A0564, Dako); rabbit anti-Glut-2, 1:500 (#07-1402, Millipore); rat anti-BrDU, 1:200 (#MCA2060, AbD Serotec); mouse anti-proinsulin, 1:200 (clone GS9A8, Developmental Studies Hybridoma Bank, University of Iowa); and mouse anti-Ki76, 1:200 (#550609, BD Biosciences). Primary antibodies were detected with Alexa-488, Alexa-555 and Alexa-633 conjugated secondary antibodies (#A11029, #A11034, #A11073, #A21428, #A21435, #A21422, #A21105, Invitrogen) or biotinylated anti-rabbit (#111-065-003, Jackson ImmunoResearch) and anti-guinea pig (#BA-7000, Vector Labs) antibodies, all used at 1:200 dilutions. For islet mass quantification, sections 100 μm apart were stained with anti-insulin antibody and total islet area quantified as a percent of total pancreatic area. Bright field images were acquired using a Zeiss Axio Imager D1 microscope. Zeiss Axioscope2 wide field and Zeiss ApoTome microscopes were used to visualize fluorescence. Unless otherwise noted, all photomicrographs shown are representative of at least three independent samples of the indicated genotype.

**BrdU incorporation.** BrdU (Sigma B9285) was reconstituted at 10 mg per ml in saline solution and administered intra-peritoneally at 50 μg per g of body weight. Mice were euthanized after 6 h, pancreata were isolated, fixed in Z-Fix for 12–16 h at 4 °C, and processed for paraffin embedding. Pancreatic sections were de-paraffinized, rehydrated and treated for 20 min with 2 N HCl at 37 °C, followed by washes with water and PBS. Slides were then processed for immunostaining as follows—sections were incubated in blocking reagent (1% BSA in PBS) for 30 min, followed by incubation with the appropriate primary antibodies in the blocking reagent overnight at 4 °C. Slides were washed three times in PBS for 5 min each, followed by incubation for 30 min at room temperature in the appropriate secondary antibodies in blocking reagent. Slides were washed again in PBS (three times for 5 min each) and mounted using VectaShield HardSet mounting medium with DAPI (Vector Laboratories)[47]. BrdU-positive cells were counted and represented as a percent of insulin positive cells and per islet.

**Western blotting.** For westerns, isolated islets or INS-1 cells grown in six well plates were homogenized in RIPA buffer and resolved on SDS-PAGE. Primary antibodies used were rabbit anti-c-Myc, 1:1000 (#5605, Cell Signaling, Boston, MA) and mouse anti-Gapdh, 1:5000 (#sc-32233, Santa Cruz). Secondary antibodies were anti-rabbit IRDye 800CW (#827-08365, Odyssey) and anti-mouse IRDye 680LT (#827-11080, Odyssey). Western analysis for cell cycle proteins on islets isolated from mice was carried out using the following antibodies—anti-Cdk1, 1:1000 (#9112, Cell Signaling Technologies), anti-Cdk2, 1:500 (#163, Santa Cruz Biotechnology), anti-Cdk4, 1:1000 (#260, Santa Cruz Biotechnology), anti-Cdk6, 1:500 (#3126, Abcam Inc.), anti-Cyclin A, 1:500 (#4710, Sigma), anti-Cyclin D3, 1:500 (#28283, Abcam Inc.), anti-Cyclin E, 1:500 (#481, Santa Cruz Biotechnology), anti-tubulin, 1:2000 (Calbiochem) and anti-actin, 1:2000 (Sigma).[7]

**Quantitative PCR and gene expression array.** RNA isolation was carried out using the RNeasy kit (Qiagen) as per manufacturer's instructions. cDNA preparation was carried out using the SuperScript III First Strand synthesis kit (Thermo Fisher Scientific), and quantitative PCR (qPCR) were performed using Fast SyBr green mix (Thermo Fisher Scientific) as per manufacturer's instructions[48]. RNA expression of target genes was normalized to Cyclophilin A expression for mouse samples. Fast SyBr green was used for all qPCR reactions. One control was set to 1 and all other controls and test samples were normalized to that sample. Primer sequences are included in Supplementary Table 1.

**RNA sequencing and data analysis.** RNA-seq data was analyzed under the Aroma framework (http://CRAN.R-project.org/package = aroma.core) in R/Bioconductor[49] (http://www.R-project.org/). The data were annotated with ENSEMBL Genome Reference Consortium Mouse Reference 38 release 79 (GRCm38.79) assembly, identifying each feature (tag) by its ENSEMBL gene ID. The alignment for each sample was read using the TopHat v2 aligner in aroma.seq package (https://github.com/HenrikBengtsson/aroma.seq). Groups were compared by performing exact test for negative-binomially distributed count data using edgeR package[50]. A nominal min-value cut-off of $10^{-6}$ was used to determine significantly differentially expressed genes. The RNA-seq data have been deposited in NCBI GEO under the accession code GSE107617.

**Pyronin Y staining.** Islets were dissociated in Dissociation buffer for 30 min at 37 °C with shaking every 10 min. The cells were passed through a 40 μm filter, spun down, and incubated with 1 ml FACS buffer with 28 mM glucose and 1 μg per ml Hoechst 33342 (HO) for 45 min at 37 °C. Subsequently, cells were spun down, and fresh 1 ml FACS buffer with 2.8 mM glucose was added with 1 μg per ml HO and 1 μg per ml Pyronin Y (PY) for 40 min at 37 °C, covered with foil. The cells were put on ice until run on the analyzer.

**hESC culture and differentiation.** hESC-derived β-like cells were generated using our recently published differentiation approach with improvements at the last two stages[16]. Clusters containing β-like cells were dissociated at day 16 by Accumax

treatment, re-aggregated using Aggrewells (StemCell Technologies) in the presence of 5 μM ROCK inhibitor and incubated for 3 days with adenoviruses. Three days later, clusters were dissociated, fixed for ~15 min in 4% PFA and stained with direct conjugated antibodies against human C-peptide (#05-1109, Millipore, conjugated using a commercial available antibody labeling kit (Invitrogen)) and human Ki67 (#550609, BD Bioscience). Flow cytometry was used to analyze Ki67 co-expression in c-peptide positive cells.

**Chromatin immunoprecipitation.** Islets were dispersed into single cells in Cell Dissociation Buffer (Gibco) and subsequent ChIP was performed using the True MicroChIP kit (Diagenode). Briefly, cells were fixed in 1% formaldehyde (28906, Thermo Fisher) for 10 min at RT and quenched with 125 mM glycine. Fixed cells were resuspended in lysis buffer and sheared using the Bioruptor Pro (Diagenode) for 2 × 10 cycles of 30 s "ON"/30 s "OFF" at high power setting to obtain fragments of 150–500 bp. 1/20 of sheared chromatin was kept aside as input and the remaining chromatin was incubated overnight with 1 μg rabbit IgG antibody (provided in kit) or 3 μg anti c-Myc antibody (ab56, Abcam). Antibody-chromatin complexes were pulled down using Protein A-coated magnetic beads and de-crosslinked for 4 h. The DNA was purified using MicroChIP DiaPure columns (Diagenode) and quantitative PCR performed using the FastStart Universal SYBR Green Master (Roche). Percent input values were calculated as $100 \times 2^{(\text{Ct[adjusted input]}-\text{Ct[IP]})}$. Primers were designed to amplify regions surrounding the canonical Myc binding motif (CACGTG) found up to 10 kb upstream of the transcriptional start site of the genes of interest. Primer sequences are included in Supplementary Table 1.

**Human chromatin immunoprecipitation data analyses.** Raw reads from Arda et al.[31] were downloaded from NCBI GEO [GSE79469]. The raw reads were aligned with hg19 assembly by using bowtie for Illumina. MACS2 was used for peak calling. Peaks were annotated by peak2gene for their locations relative to adjacent genes. Peaks were visualized by EaSeq [reference: http://www.nature.com/nsmb/journal/v23/n4/full/nsmb.3180.html].

**Statistics.** Data are presented as mean ± SD or mean ± SEM and subjected to two-tailed unpaired t test. A p value of 0.05 or lower was considered to be significant.

**Data availability.** All relevant data pertaining to this study are available within the article as well as Supplementary Information files, and from the corresponding author upon request.

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

## Acknowledgements

We thank Drs. Jennifer Liu, Gopika Nair, and Audrey Parent for critical reading of the manuscript, Debbie Ngow for assistance with tissue processing, and Larry Ackerman for assistance with electron microscopy. We especially acknowledge Drs. Yuval Dor and Agnes Klochendler for their valuable insights and helpful discussions during the course of this study. This work was supported by grants from the Leona M. and Harry B. Helmsley Charitable Trust (2012PG-T1D016 to M.H.), by the JDRF (17-2011-598 to A.F.S., M.H. and D.K.S.; SRA 17-2011-59 and SRA 17-2015-62 to A.F.S.; Postdoctoral Fellowship 3-2012-266 to H.A.R.), by the NIH/NIDDK (R-01 DK 55023, R-01 015015, UC4 DK104211 and P-30 DK 0205241 to A.F.S.), and by a Richard G. Klein Fellowship (to H.A.R.). We thank the UCSF Diabetes and Endocrinology Research Center (DERC) the Microscopy Core and Islet Isolation Core (P30 DK63720) for islet isolation. The UCSF Cancer Center Support Grant P30 CA082103 provided support for administration and infrastructure for the UCSF Comprehensive Cancer Center.

## Author contributions

S.P. and M.H. conceived the study. S.P. designed and conducted the experiments and wrote the paper. H.A.R. conducted experiments and edited the paper. N.R., L.L., and E.K. F. conducted experiments; R.R and H.B. conducted bioinformatics analyses. D.K.S., A.F. S., and M.H. edited the paper and participated in discussions during the design of the study and the interpretation of the data.

## Additional information

**Competing interests:** The authors declare no competing financial interests.

