## [Peer Review File · Nature Communications]

Reviewer #1 (Remarks to the Author):

This study by Puri et al investigates whether c-Myc gain- and loss-of-function has an impact on beta cell proliferation and maturation. The starting hypothesis is that proliferating beta cells are less mature than non-proliferative ones. Chemical and siRNA-mediated inhibition or knock-down in INS-1 cells decrease proliferation and increase maturation marker expression and glucose-stimulated insulin secretion. In contrast, Insulin promoter-driven c-Myc overexpression in beta cells increases beta cell replication and mass in vivo, but impairs glucose regulation. Beta cells overexpressing c-Myc show more immature granules, decreased levels of PC1/3, increased basal proinsulin and global changes in the gene expression profile, leading the author to conclude that replicating beta cells are less mature than quiescent ones. Taken together, this is an interesting study that is both of mechanistic and therapeutic relevance.

I have one general and a few more specific comments, which should be addressed before publication. [REDACTED]

Specific comments:

1. In Figure 1 the authors describe the effect of c-Myc chemical inhibition and siRNA-mediated knock-down in INS-1 cells. How specific is the Myc inhibitor at the very high concentration of 40 μ M and which potential off targets does it have? Why is the effect of the siRNA knock-down less pronounced? According to the quantification of the WB shown in Supplementary Fig. 1c the knock-down is very efficient, but the loading of the gel is very bad, as shown by the loading control. Is the knock-down efficiency overestimated? Or has the chemical inhibition additional side effects?
2. In Figure 2 c-Myc is delivered by adenovirus to hESC-derived insulin-producing cells. What was the efficiency of adenovirus transduction? Would it not make sense to present Ki67+/C-peptide+ and EdU+/C-peptide+ cells separately?
3. In Figure 3 and Supplementary Fig. 3 the authors show that Ins-c-Myc overexpression leads to increased basal insulin, increased beta cell mass, and hypoglycemia. They argue that beta cells poised to proliferate have high insulin secretion and impaired secretory response. How can the authors rule out that they have generated a neomorphic phenotype due to c-Myc GOF? What is the phenotype of a beta cell that is naturally dividing? If beta cells anyway do not divide in human islets, what is the relevance?
4. In Figure 4 the authors describe the ultrastructural and immature feature of the c-Myc expressing beta cells. It would be nice if they explain the phenotypes they observe (immature granules, PC1/3 downregulation) in molecular terms. Does c-Myc directly or indirectly regulate PC1/3 expression, secretory pathway components, maturation and cell cycle programs (see next comment).
5. In Figure 5 and 6 the authors use mRNA profile and published ChIP seq data sets to underpin c-Myc molecular function in islet cell proliferation and maturation. For me it is not clear what are the direct effects of c-Myc overexpression and what is indirect? For example, Pdx1, Nkx6.1, MafA, and NeuroD1 are all reduced in transgenic samples. Where they found in the mRNA profiling and then confirmed on qPCR level and then checked if c-Myc binds to their enhancers or promoters? It would be nice that for some genes e.g. Pc1/3 the authors explain what might be direct and indirect effects of c-Myc overexpression. Does c-Myc primarily change the cell-cycle state and downstream of this maturation programs change or does c-Myc also directly regulate maturation programs?
6. The discussion should include a more mechanistic discussion about proliferating cells in homeostasis and pregnancy (which the authors touch on) and how this compares to changes in gene expression the authors see in their c-Myc model. Rieck and Kaesnter have an excellent research paper and review on pregnancy and the factors driving compensatory beta cell proliferation.

Reviewer #2 (Remarks to the Author):

This truly important study deals in a very elegant way with a long-standing issue in pancreatic beta-cell biology, the suggested inverse relationship between beta-cell functionality (as a sign of maturity) and proliferative capacity. Using well designed models the authors convincingly demonstrate that the cell cycle regulator c-Myc controls this balance. It could be argued that this is an condition that has limited physiological relevance, but results from experiments in pregnant mice supports the view that c-Myc acts in concert with known factors stimulating beta-cell proliferation in real life.

The statistical evaluation is fairly straightforward and well performed with adequate number of observations. The level of methodological detail seems sufficient for other reserachers to undertand the how to reproduce the findings independently.

Comments:

1.

During embryonal development pancreatic beta-cells (also in humans) have been reported not to be responsive to glucose. This capacity develops after birth, starting with a monophasic insulin secretion response that gradually develops into the mature biphasic pattern. Embryonal beta-cells are, however extremely responsive to glucagon or to pharmacological agents that increase cAMP. It would substantially increase or understanding of how the c-Myc induced immaturity relates to normal pancreatic development if the capacity of the alternative early secretagogue glucagon to induce insulin secretion was investigated in Ins-c-Myc islets. It would also be of great interest to investigate the effects of GLP-1 and cAMP-elevating pharmacological agents e.g. forskolin.

2.

The concept of dedifferentiation remains somewhat controversial, but it would also be of great general interest if proposed patterns of transcription factors and other markers of dedifferentiation were investigated in Ins-c-Myc islets (the authors have most of those results already) and discussed. This would of course not serve as the final say on dedifferentiation, but would help to understand what type of immature state Ins-c-Myc islets represent.

Reviewers' comments:

Reviewer #1 (Remarks to the Author):

This study by Puri et al investigates whether c-Myc gain- and loss-of-function has an impact on beta cell proliferation and maturation. The starting hypothesis is that proliferating beta cells are less mature than non-proliferative ones. Chemical and siRNA-mediated inhibition or knock-down in INS-1 cells decrease proliferation and increase maturation marker expression and glucose-stimulated insulin secretion. In contrast, Insulin promoter-driven c-Myc overexpression in beta cells increases beta cell replication and mass in vivo, but impairs glucose regulation. Beta cells overexpressing c-Myc show more immature granules, decreased levels of PC1/3, increased basal proinsulin and global changes in the gene expression profile, leading the author to conclude that replicating beta cells are less mature than quiescent ones. Taken together, this is an interesting study that is both of mechanistic and therapeutic relevance.

We thank the reviewer for pointing out the general relevance and importance of our study, and appreciating the “mechanistic and therapeutic relevance” of the findings. We agree that our observations inform the research currently focusing on beta cell fate, function, and expansion.

I have one general and a few more specific comments, which should be addressed before publication. [REDACTED].

[REDACTED]

Specific comments:

1. In Figure 1 the authors describe the effect of c-Myc chemical inhibition and siRNA-mediated knock-down in INS-1 cells. How specific is the Myc inhibitor at the very high concentration of 40 μM and which potential off targets does it have? Why is the effect of the siRNA knock-down less pronounced? According to the quantification of the WB shown in Supplementary Fig. 1c the knock-down is very efficient, but the loading of the gel is very bad, as shown by the loading control. Is the knock-down efficiency overestimated? Or has the chemical inhibition additional side effects?

Although an effective inhibitor of the c-Myc/Max interaction, 10058-F4 has been shown to also block N-Myc/Max interactions effectively in neuroblastoma cells, with a K_D of 41.9 μM as compared to a K_D of 39.7 μM for c-Myc (Zirath et al., 2013, Muller et al., 2014). Therefore, a more dramatic reduction in total Myc activity may explain the profound effect on INS-1 cells. The siRNA, however, was only directed to c-Myc, and might reveal milder effects.

2. In Figure 2 c-Myc is delivered by adenovirus to hESC-derived insulin-producing cells. What was the efficiency of adenovirus transduction? Would it not make sense to present Ki67+/C-peptide+ and EdU+/C-peptide+ cells separately?

Unfortunately, we did not quantify the efficiency of the adenovirus transduction. As the reviewer suggested, we have simplified Figure 2h to represent the %Ki67-positive cells in the C-peptide positive cells after the adenoviral infection.

3. In Figure 3 and Supplementary Fig. 3 the authors show that *Ins-c-Myc* overexpression leads to increased basal insulin, increased beta cell mass, and hypoglycemia. They argue that beta cells poised to proliferate have high insulin secretion and impaired secretory response. How can the authors rule out that they have generated a neomorphic phenotype due to c-Myc GOF?

It is difficult for us to categorically disprove that the cells with c-Myc GOF have adopted a neomorphic phenotype (if we interpret 'neomorphic' as a new gene activity of c-Myc that causes these changes in the beta cell). Having said this, there is a vast body of literature that places c-Myc as an essential regulator of cell cycle machinery as well as differentiation genes. We have correlative evidence that c-Myc is present at developmental time points when beta cells are replicating, strongly implicating a role for c-Myc in 'normal/unperturbed' post-natal beta cell expansion. More definitive proof of c-Myc involvement comes from *in vivo* reduction of c-Myc in beta cells that leads to reduced cellular proliferation. Our *in vitro* data show that the functional changes that we observe *in vivo* are in fact reversible and can be rescued with reduction in c-Myc expression. All these data suggest a role for c-Myc in promoting beta cell proliferation and modifying beta cell identity in a reversible manner.

What is the phenotype of a beta cell that is naturally dividing?

Our studies point to functional immaturity or reduced functional capacity as the critical phenotype of a dividing cell. Naturally dividing beta cells are infrequent, and it has proven difficult to conduct meaningful physiological studies on them. Our comparison of genes that are upregulated in c-Myc GOF cells and non-replicating postnatal cells demonstrates that c-Myc primes a cell to divide, with the additional consequence of immature function. Three pieces of evidence that we provide in this study – 1) the poor secretory capacity at early postnatal stages, 2) increased Myc levels under physiological

levels at those stages, and 3) overlap between increased Myc target genes and those early physiological states – lead us to conclude that immature islets are primed to replicate, with only a fraction of cells actively replicating. This model for the first time has allowed us to dissect such functional differences between the replicating and non-replicating beta cell populations within the developing islet. Ideally, it would be beneficial to individually test the secretory capacity of proliferating and non-proliferating beta cells that are present in the early postnatal states in the absence of genetic manipulation, and conduct a comparison with the *Ins-c-Myc* model, however, such experiments, albeit interesting, are beyond the scope of the current manuscript.

If beta cells anyway do not divide in human islets, what is the relevance?

We agree with the reviewer that beta cell proliferation is severely curbed in human adults. Early in life (<5 years), however, there is a period of significant beta cell expansion, and thus it is likely that c-Myc plays a role at those stages. Therefore, beta cells can proliferate after birth, and trying to understand the mechanisms by which this process is regulated could have profound implications for treatment if such activities could be reinstated in adult beta cells. In addition, such information would help to test whether beta cells that have been harvested from cadaveric donors could be expanded. These cells may be coaxed through various means to multiply, however, it is important to keep in mind that such effects need to be reversed so that the beta cell can re-acquire a more mature, functional phenotype. Transplantation of beta cells that have been engineered to divide either *in vivo* or *in vitro* could prove disastrous in terms of the ability of these cells to regulate glucose levels appropriately. Thus, we believe the relevance of our work is at least 2-fold - identifying a novel, physiological regulator of beta cell expansion, fate and function, and the observation that replication-competent beta cells are suboptimal in function.

4. In Figure 4 the authors describe the ultrastructural and immature feature of the c-Myc expressing beta cells. It would be nice if they explain the phenotypes they observe (immature granules, PC1/3 downregulation) in molecular terms. Does c-Myc directly or indirectly regulate PC1/3 expression, secretory pathway components, maturation and cell cycle programs (see next comment).

In Figure 5 and 6 the authors use mRNA profile and published ChIP seq data sets to underpin c-Myc molecular function in islet cell proliferation and maturation. For me it is not clear what are the direct effects of c-Myc overexpression and what is indirect? For example, Pdx1, Nkx6.1, MafA, and NeuroD1 are all reduced in transgenic samples. Were they found in the mRNA profiling and then confirmed on qPCR level and then checked if c-Myc binds to their enhancers or promoters? It would be nice that for some genes e.g. Pc1/3 the authors explain what might be direct and indirect effects of c-Myc overexpression. Does c-Myc primarily change the cell-cycle state and downstream of this maturation programs change or does c-Myc also directly regulate maturation programs?

We appreciate the reviewer's comments about the direct versus indirect effects of Myc overexpression. As the reviewer will agree, Myc binds a large proportion of the genome (assessed to be >15% of the genome in lymphoma cells, Li et al., 2003, Zeller et al., 2006). A large effort beyond the scope of this study is necessary to define exactly which of the activities of Myc are mediated via direct binding to regulatory elements of the gene in question. Indeed, we plan to pursue this line of investigation in the future to establish novel roles of Myc in beta cell fate and function. However, to address the reviewer's comments, we did attempt to identify Myc binding at some of the canonical beta cell genes, and show our data in the revised version of Fig. 4 (Fig. 4g). In short, we see Myc

binding at canonical binding sites identified in upstream regions of the genes shown in the revised figure. We evaluated regions 10kb upstream of the transcriptional start site for several beta cell genes, and identified canonical Myc binding sites (CACGTG). The locations of the binding sites for the genes including *Pcsk1*, *Pdx1*, *Neurod1*, and *Ins2*, are noted in the revised version of the manuscript. We did not find any canonical binding sites for *Nkx6.1*, *Mafa* and *Ins1* gene in the 10kb region upstream of the transcriptional start site. We believe these results are significant, as this is the first time there has been direct evidence that Myc may be a potential regulator of beta cell identity and maturation. Due to limited availability of transgenic mice, we were unable to test Myc binding at the other gene locations suggested by the reviewer.

6. The discussion should include a more mechanistic discussion about proliferating cells in homeostasis and pregnancy (which the authors touch on) and how this compares to changes in gene expression the authors see in their c-Myc model. Rieck and Kaesner have an excellent research paper and review on pregnancy and the factors driving compensatory beta cell proliferation.

We appreciate the reviewer's comment about including a more mechanistic discussion on beta cell proliferation during homeostasis and pregnancy. As the reviewer has pointed out, we conducted an analysis on beta cell expansion during pregnancy, and demonstrated that a physiological condition such as pregnancy has a significant impact on beta cells that are primed to divide. In the revised manuscript we include an updated discussion to further address the reviewer's comment.

Reviewer #2 (Remarks to the Author):

This truly important study deals in a very elegant way with a long-standing issue in pancreatic beta-cell biology, the suggested inverse relationship between beta-cell functionality (as a sign of maturity) and proliferative capacity. Using well-designed models the authors convincingly demonstrate that the cell cycle regulator c-Myc controls this balance. It could be argued that this is an condition that has limited physiological relevance, but results from experiments in pregnant mice supports the view that c-Myc acts in concert with known factors stimulating beta-cell proliferation in real life.

The statistical evaluation is fairly straightforward and well performed with adequate number of observations. The level of methodological detail seems sufficient for other researchers to understand how to reproduce the findings independently.

We very much appreciate the reviewer's assessment of our manuscript, and thank him/her for their input that allows us to strengthen the research.

Comments:

*1. During embryonal development pancreatic beta-cells (also in humans) have been reported not to be responsive to glucose. This capacity develops after birth, starting with a monophasic insulin secretion response that gradually develops into the mature biphasic pattern. Embryonal beta-cells are, however extremely responsive to glucagon or to pharmacological agents that increase cAMP. It would substantially increase our understanding of how the c-Myc induced immaturity relates to normal pancreatic development if the capacity of the alternative early secretagogue glucagon to induce insulin secretion was investigated in *Ins-c-Myc* islets. It would also be of great interest to investigate the effects of GLP-1 and cAMP-elevating pharmacological agents e.g. forskolin.*

We greatly appreciate the comments and suggestions of the reviewer. In order to assess whether the Myc GOF islets were responsive to alternate secretagogues, we isolated islets from 6-month-old control and *Ins-c-Myc* animals and carried out a GSIS in the presence of glucagon (10nM), Glp-1 (100nM), or forskolin (10μM) in 16.7 mM glucose. Our data have been included in the revised Figure 3g. We found, as before, that while control islets demonstrated strict control of glucose stimulated insulin secretion with low levels of insulin secreted under 2.8mM glucose conditions, c-Myc-expressing islets had high insulin secretion at 2.8mM. Addition of glucagon, Glp-1 or forskolin improved secretion significantly from c-Myc islets, suggesting that these secretagogues could further stimulate insulin secretion, similar to what is seen in embryonic beta cells that respond better to these secretagogues as compared to glucose. These data further strengthen our conclusion that c-Myc-expressing islets functionally mimic fetal or early postnatal stages.

2. The concept of dedifferentiation remains somewhat controversial, but it would also be of great general interest if proposed patterns of transcription factors and other markers of dedifferentiation were investigated in Ins-c-Myc islets (the authors have most of those results already) and discussed. This would of course not serve as the final say on dedifferentiation, but would help to understand what type of immature state Ins-c-Myc islets represent.

As the reviewer points out, we have included gene expression quantification of several markers including *Pdx1*, *Mafa*, *Neurod1*, *Nkx6.1*, *Ucn3*, *Glut2* and *Gck* in addition to *Ins1* and *Ins2* in Supplemental figure 4a in the original manuscript. Overall, we observe a reduction in the expression of several of these markers in the transgenic islets as compared to the control littermates. Furthermore, one of the prominent disallowed genes, *Mct1*, was significantly upregulated in the *Ins-c-Myc* islets, further supporting an immature phenotype. As the reviewer points out, although controversial, dedifferentiation within beta cells is well accepted as one mechanism of beta cell dysfunction. It is likely that significant overlaps exist between an immature cell state and a dedifferentiated state, since both are characterized by reduced expression of the canonical transcription factors and other functionally important genes in beta cells. Such overlap is also reflected in the appearance of disallowed genes, that are expressed early in life but suppressed as beta cells mature, and can be reactivated under conditions of stress later in life. We believe that chronic upregulation of c-Myc in beta cells suppresses beta cell identity, leading to dramatic downregulation of canonical beta cell gene expression and loss of function. In the revised manuscript, we further our findings by expanding our gene expression analysis to include *Nkx2.2*, *Pax6*, and *Isl1*. We find a significant reduction in the additional genes; further supporting our original data that beta cell dedifferentiation can be detected at 3 months of age. We did examine the dedifferentiation factor *Foxo1*, however, we did not see any change in its gene expression, most likely due to the early age of the mice, as *Foxo1* deletion has effects after aging or upon exposure to physiological stress. We have included the new data in a modified version of Supplemental Figure 4a. Also, we have modified the discussion to include our findings regarding beta cell dedifferentiation as recommended by the reviewer.

Reviewer #1 (Remarks to the Author):

all my concerns have been addressed.

Reviewer #2 (Remarks to the Author):

The authors have addressed the feedback in my review by adding relevant new data and adjusting the discussion. No further comments.